# Fast Iterative Hard Thresholding Methods with Pruning Gradient Computations

**Yasutoshi Ida**[1][*]   **Sekitoshi Kanai**[1]   **Atsutoshi Kumagai**[1]   **Tomoharu Iwata**[2]

**Yasuhiro Fujiwara**[2]

[1]NTT Computer and Data Science Laboratories
[2]NTT Communication Science Laboratories

## Abstract

We accelerate the iterative hard thresholding (IHT) method, which finds $k$ important elements from a parameter vector in a linear regression model. Although the plain IHT repeatedly updates the parameter vector during the optimization, computing gradients is the main bottleneck. Our method safely prunes unnecessary gradient computations to reduce the processing time. The main idea is to efficiently construct a candidate set, which contains $k$ important elements in the parameter vector, for each iteration. Specifically, before computing the gradients, we prune unnecessary elements in the parameter vector for the candidate set by utilizing upper bounds on absolute values of the parameters. Our method guarantees the same optimization results as the plain IHT because our pruning is safe. Experiments show that our method is up to 73 times faster than the plain IHT without degrading accuracy.

## 1   Introduction

The optimization problem of finding sparse parameter vectors in linear regression models is a crucial problem that crosses a wide range of fields, including feature selection [20, 35], sparse coding [32], dictionary learning [33], and compressed sensing [6]. To obtain the sparse parameter vector, the parameter vector is often constrained to have $k$ nonzero elements. Among a huge number of algorithms that have been developed for this problem [25, 29, 14, 27, 10, 19, 11, 20], iterative hard thresholding (IHT) methods [6] are practical methods based on gradient descent methods because they have almost no overhead over the plain gradient descent method [2].

A procedure of IHT consists of two main parts in an iteration. First, it updates the parameter vector in accordance with the gradient descent method. Then, the parameter vector is projected onto a feasible set being a set of sparse parameter vectors. In the projection, IHT uses a hard thresholding operator that selects the $k$-largest elements in terms of magnitude of the parameter vector, and the other elements are set to zero. IHT repeats this procedure until the stopping criterion is satisfied.

When a design matrix of the linear regression model is an $m$-by-$n$ matrix, the length of the parameter vector is $n$, and gradient computations of IHT require $\mathcal{O}(mn)$ or $\mathcal{O}(n^2)$ time for each iteration. Specifically, IHT suffers from the increase in processing time for large $m$ and $n$. Taking feature selection as an example, IHT slows down for large datasets because $m$ and $n$ correspond to the numbers of samples and features in the dataset, respectively. The gradient computations are much more computationally expensive than the projection of the parameter vector because the projection only requires $\mathcal{O}(n \log k)$ time if it uses a heap. Therefore, the gradient computations are dominant in the overall procedure of IHT, and we need to reduce the cost to raise the efficiency of IHT.

---

[*]Corresponding author: yasutoshi.ida@ieee.org

38th Conference on Neural Information Processing Systems (NeurIPS 2024).

This paper proposes fast IHT that safely prunes unnecessary gradient computations. Before computing the gradients for each iteration, the proposed method efficiently constructs a candidate set whose elements correspond to indices of the $k$-largest elements in terms of magnitude of the parameter vector. When constructing the candidate set, our method prunes indices that are clearly not included in the candidate set. To identify such indices, we compute upper bounds of absolute values for the elements of the parameter vector. If the upper bound is smaller than a threshold, the index is not included in the candidate set. The threshold is automatically determined by leveraging lower bounds of absolute values for the elements of the parameter vector. Since the computation cost of the upper and lower bounds is $\mathcal{O}(n)$ time, we can efficiently construct the candidate set. By updating only the parameters corresponding to the candidate set, we can prune unnecessary gradient computations. Our method guarantees the same optimization results as the plain IHT because it safely prunes unnecessary computations. In addition, our method does not need additional hyperparameter tuning. Experiments demonstrate that our method is up to 73 times faster than the plain IHT while maintaining accuracy on feature selection tasks.

## 2 Related Work

In [4, 30], the authors utilized a double-overrelaxation approach to improve the convergence speed of IHT. They use two relaxation steps for the parameter vector, which are similar to the momentum of Nesterov's method [28]. In [8, 24], the authors introduced the momentum to IHT inspired by the fast iterative shrinkage thresholding algorithm (FISTA) [3]. While FISTA uses the momentum with a soft thresholding operator, Cevher [8] uses it with the hard thresholding operator. This Accelerated IHT (AccIHT) has substantial theoretical and empirical improvement over the plain IHT [23].

While the previous methods have reduced the number of iterations to accelerate IHT as described above, to the best of our knowledge, there are no papers on reducing the computation cost per iteration of IHT. This paper aims to fill this gap based on the pruning strategy. For convex and some nonconvex regularization, working set algorithms are used to reduce the cost of solvers [7, 21, 26, 31]. They solve a growing sequence of subproblems that are restricted to a small subset of parameters during optimization. In [12, 13, 22, 15, 16, 18, 17], the authors reduced the cost by skipping unnecessary parameter updates for coordinate descent with sparsity-inducing norms. However, since these methods are tailored for coordinate descent or the soft thresholding operator, they cannot be used for IHT, which selects $k$ elements from the parameter vector by using the hard thresholding operator.

## 3 Preliminary

**Notation.** We denote scalars, vectors, and matrices with lower-case, bold lower-case, and bold upper-case letters, *e.g.*, $x$, $\boldsymbol{x}$ and $\boldsymbol{X}$, respectively. Given a matrix $\boldsymbol{X}$, we denote its $i$-th row by $\boldsymbol{X}_i$. Given a vector $\boldsymbol{x} \in \mathbb{R}^m$, we denote its $i$-th element by $\boldsymbol{x}_i$, and we call $i$ index. $\|\cdot\|_2$ is the $\ell_2$ norm. $\|\boldsymbol{x}\|_0$ is $|\{i \in \{1, ..., m\}|x_i \neq 0\}|$ and represents the number of nonzero elements in $\boldsymbol{x}$. $\boldsymbol{0} \in \mathbb{R}^m$ is the $m$-dimensional vector whose elements are zeros. $\boldsymbol{I}$ represents the identity matrix. $\text{supp}(\boldsymbol{x})$ is the function that returns the indices of nonzero elements in $\boldsymbol{x}$.

### 3.1 Problem Setting

Let $\boldsymbol{X} \in \mathbb{R}^{m \times n}$ be an input matrix (design matrix), $\boldsymbol{y} \in \mathbb{R}^m$ be a set of continuous responses, and $\boldsymbol{\theta} \in \mathbb{R}^n$ be a parameter vector of a linear regression model. To find a sparse parameter vector of the model, we consider the following optimization problem [5]:

$$\min_{\boldsymbol{\theta} \in \mathbb{R}^n} \tfrac{1}{2}\|\boldsymbol{y} - \boldsymbol{X}\boldsymbol{\theta}\|_2^2 \quad \text{subject to} \quad \|\boldsymbol{\theta}\|_0 \leq k. \tag{1}$$

In the above problem, the number of nonzero elements in the parameter vector, $\|\boldsymbol{\theta}\|_0$, is constrained by $k \in \{1, ..., n\}$. Here, we will let $f(\boldsymbol{\theta}) = \tfrac{1}{2}\|\boldsymbol{y} - \boldsymbol{X}\boldsymbol{\theta}\|_2^2$ for simplicity.

### 3.2 Iterative Hard Thresholding

IHT is the practical algorithm for Problem (1) [5, 2]. It repeatedly performs the following iteration:

$$\boldsymbol{z}^t = \boldsymbol{\theta}^t - \eta \nabla f(\boldsymbol{\theta}^t) = \boldsymbol{\theta}^t - \eta \boldsymbol{X}^\top(\boldsymbol{X}\boldsymbol{\theta}^t - \boldsymbol{y}) = (\boldsymbol{I} - \eta \boldsymbol{X}^\top \boldsymbol{X})\boldsymbol{\theta}^t + \eta \boldsymbol{X}^\top \boldsymbol{y}, \tag{2}$$

$$\boldsymbol{\theta}^{t+1} = H_k(\boldsymbol{z}^t), \tag{3}$$

---
**Algorithm 1** Iterative Hard Thresholding
---
1: **Input:** sparsity level $k$, step size $\eta$
2: **Initialization:** $\boldsymbol{\theta}^1 \leftarrow \mathbf{0}, t \leftarrow 1$
3: **repeat**
4: $\quad \mid \quad \boldsymbol{z}^t \leftarrow \boldsymbol{\theta}^t - \eta \nabla f(\boldsymbol{\theta}^t);$ $\qquad\qquad\qquad\qquad$ ▷ Performing gradient descent method
5: $\quad \mid \quad \boldsymbol{\theta}^{t+1} \leftarrow H_k(\boldsymbol{z}^t); t \leftarrow t + 1;$ $\qquad$ ▷ Selecting the $k$-largest elements in magnitude of $\boldsymbol{z}^t$
6: **until** a stopping criterion is met
---

where $\eta > 0$ is the step size, $\boldsymbol{\theta}^t$ is the parameter vector at the $t$-th iteration. $H_k(\boldsymbol{z}^t)$ is the hard thresholding operator that selects the $k$-largest elements in the magnitude of $\boldsymbol{z}^t$ and sets the other elements to zero. The selection requires $\mathcal{O}(n \log k)$ time if it uses a heap. The pseudocode is described in Algorithm 1. See [5, 14, 6, 8, 4, 20, 23, 2] for theoretical discussions of IHT.

From Equation (2) and Algorithm 1, computing gradients $\nabla f(\boldsymbol{\theta}^t)$ clearly dominates the other cost. In Equation (2), we use the second equation or the third one to compute $\boldsymbol{z}^t$. They require $\mathcal{O}(mn)$ and $\mathcal{O}(n^2)$ times in every iteration, respectively[2]. Therefore, IHT incurs high computation cost when $\boldsymbol{X} \in \mathbb{R}^{m \times n}$ is large.

## 4 Proposed Algorithm

This section describes our algorithm that reduces the computation cost per iteration in IHT.

### 4.1 Main Idea

The bottleneck of IHT is the gradient computation to obtain $\boldsymbol{z}^t$ of Equation (2): it requires $\mathcal{O}(mn)$ or $\mathcal{O}(n^2)$ time per iteration. Therefore, we reduce the cost by pruning unnecessary elements in $\boldsymbol{z}^t$ before computing the gradients. For the pruning, we introduce a candidate set $\mathcal{D}^t$ such that $|\mathcal{D}^t| = k$ for the $t$-th iteration. This set maintains indices of nonzero elements of the parameter vector during optimization. In other words, the candidate set contains indices of the $k$-largest elements in terms of magnitude of the parameter vector. Before computing Equation (2), we quickly check whether indices of elements in $\boldsymbol{z}^t$ are included or not in $\mathcal{D}^{t+1}$. If an index $j$ is not included in $\mathcal{D}^{t+1}$, we can prune $\boldsymbol{z}_j^t$ and skip the corresponding computation of Equation (2) including the gradient computation.

The point is that our method can efficiently perform the above checking procedure. Specifically, our method utilizes $\overline{\boldsymbol{z}}_j^t$, which is an upper bound of $|\boldsymbol{z}_j^t|$. Since the computation of the upper bound does not include the gradient computation, it only requires $\mathcal{O}(n)$ time for all the elements in $\overline{\boldsymbol{z}}^t$. For the checking procedure, after initializing $\mathcal{D}^t$ appropriately, our method finds a threshold for the pruning by utilizing $\underline{\boldsymbol{z}}_j^t$, which is a lower bound of $|\boldsymbol{z}_j^t|$. Then, if $\overline{\boldsymbol{z}}_j^t$ is smaller than the threshold for $j \notin \mathcal{D}^t$, the index $j$ is not included in $\mathcal{D}^{t+1}$. We describe the details in the next section.

### 4.2 Upper Bound and Candidate Set

This section introduces candidate set $\mathcal{D}^t$ and its updating method. Since we need upper bound $\overline{\boldsymbol{z}}_j^t$ to efficiently update $\mathcal{D}^t$ as described in Section 4.1, we first define $\overline{\boldsymbol{z}}_j^t$ as follows:

**Definition 1** *Let $t^*$ be $1 \leq t^* < t$ in Algorithm 1. Then, $\overline{\boldsymbol{z}}_j^t$ at the $t$-th iteration in Algorithm 1 is computed as follows:*

$$\overline{\boldsymbol{z}}_j^t = |\boldsymbol{G}_j \boldsymbol{\theta}^{t^*} + \eta(\boldsymbol{X}^\top \boldsymbol{y})_j| + \|\boldsymbol{G}_j\|_2 \|\boldsymbol{\theta}^t - \boldsymbol{\theta}^{t^*}\|_2, \tag{4}$$

*where $\boldsymbol{G} = \boldsymbol{I} - \eta \boldsymbol{X}^\top \boldsymbol{X}$.*

We note that $\boldsymbol{G}$ and $\boldsymbol{X}^\top \boldsymbol{y}$ are precomputed only once before entering the optimization, and $t^*$ is automatically decided as described in Section 4.4. $\overline{\boldsymbol{z}}_j^t$ has the following property:

---

[2]If $\boldsymbol{I} - \eta \boldsymbol{X}^\top \boldsymbol{X}$ and $\boldsymbol{X}^\top \boldsymbol{y}$ are precomputed, the cost is $\mathcal{O}(n^2)$ time. If the precomputation is not performed, the cost is $\mathcal{O}(mn)$ time because we first compute $\boldsymbol{h} = \boldsymbol{X} \boldsymbol{\theta}^t - \boldsymbol{y}$ at $\mathcal{O}(mn)$ time, then compute $\boldsymbol{X}^\top \boldsymbol{h}$ at $\mathcal{O}(mn)$ time. See the Appendix for a discussion of IHT with sparse matrices.

**Lemma 1 (Upper bound)** *When $\overline{z}_j^t$ is computed by Equation (4), we have $\overline{z}_j^t \geq |z_j^t|$.*

Lemma 1 is derived from the triangle inequality and the Cauchy–Schwarz inequality. It guarantees that $\overline{z}_j^t$ is the upper bound of $|z_j^t|$. The computation cost of the upper bound is as follows:

**Lemma 2 (Computation cost of upper bound)** *The computation cost of Equation (4) for all $j \in \{1, ..., n\}$ at the $t$-th iteration is $\mathcal{O}(n)$ time given $\boldsymbol{G}$, $\boldsymbol{X}^\top \boldsymbol{y}$ and $|\boldsymbol{G}_j \boldsymbol{\theta}^{t^*} + \eta(\boldsymbol{X}^\top \boldsymbol{y})_j|$.*

Lemma 2 shows that the upper bound at the $t$-th iteration requires only $\mathcal{O}(n)$ time if $\boldsymbol{G}$ and $\boldsymbol{X}^\top \boldsymbol{y}$ are precomputed and $|\boldsymbol{G}_j \boldsymbol{\theta}^{t^*} + \eta(\boldsymbol{X}^\top \boldsymbol{y})_j|$ is computed at the $t^*$-th iteration.

Next, we define candidate set $\mathcal{D}^t$ as follows:

**Definition 2 (Candidate set)** *Let $\mathcal{I} = \{1, ..., n\}$ be a set of all the indices in the parameter vector $\boldsymbol{\theta} \in \mathbb{R}^n$. Suppose that $\mathcal{D}^t \subset \mathcal{I}$ is a set such that $|\mathcal{D}^t| = k$ at the $t$-th iteration in IHT where $t > 1$, and initialized as $\mathcal{D}^t = \mathrm{supp}(\boldsymbol{\theta}^t)$ at the beginning of the iteration. Then, we call $\mathcal{D}^t$ the candidate set.*

$\mathcal{D}^t$ has indices that are candidates for $\mathrm{supp}(\boldsymbol{\theta}^{t+1})$ in IHT. Since $\mathcal{D}^t$ is initialized as $\mathcal{D}^t = \mathrm{supp}(\boldsymbol{\theta}^t)$ at the beginning of the iteration in Definition 2, we need to update $\mathcal{D}^t$ to $\mathcal{D}^{t+1}$ so that it matches $\mathrm{supp}(\boldsymbol{\theta}^{t+1})$. Although we can update $\mathcal{D}^t$ such that $\mathcal{D}^{t+1} = \mathrm{supp}(H_k(\boldsymbol{z}^t))$ by computing Equations (2) and (3), they include the gradient computation that requires $\mathcal{O}(mn)$ or $\mathcal{O}(n^2)$ time.

To efficiently update $\mathcal{D}^t$ to $\mathcal{D}^{t+1}$, we utilize the upper bound $\overline{z}_j^t$ of Equation (4) for $j \notin \mathcal{D}^t$. By using $\overline{z}_j^t$, we can identify unnecessary elements in $\boldsymbol{z}^t$ that are clearly not included in $\mathcal{D}^{t+1}$ as follows:

**Lemma 3 (Pruning unnecessary elements)** *Suppose that $\overline{z}^t$ is computed by using Equation (4), and the candidate set $\mathcal{D}^t$ is initialized as described in Definition 2 at the beginning of the iteration for $t > 1$. Let $z_{i_{\min}}^t$ be $z_i^t$ having the minimum $|z_i^t|$ in all $i \in \mathcal{D}^t$, and $i_{\min}$ be the index. Then, if $|z_{i_{\min}}^t| \geq \overline{z}_j^t$ holds for $j \notin \mathcal{D}^t$, $j$ is not included in $\mathcal{D}^{t+1}$.*

From Lemma 3, when we check whether $j \notin \mathcal{D}^t$ is included or not in $\mathcal{D}^{t+1}$, we do not need to compute the gradient corresponding to $z_j^t$ if $|z_{i_{\min}}^t| \geq \overline{z}_j^t$ holds. Although we need to compute the gradients to find $z_{i_{\min}}^t$ in the initial $\mathcal{D}^t$, the cost is relatively small because the cardinality of $\mathcal{D}^t$ is usually small as $|\mathcal{D}^t| = k \ll n$.

Algorithm 2 is the pseudocode of updating the candidate set. It first copies $\mathcal{D}^t$ to $\mathcal{D}^{t+1}$ (line 3). Lines 4–13 check whether the computation of $z_j^t$ can be pruned or not by following Lemma 3. If the computation is pruned (line 5), we skip the computation of $z_j^t$ including the gradient computation (line 6). If the computation is not pruned (line 7), line 8 computes $z_j^t$. If $|z_{i_{\min}}^t| < |z_j^t|$ holds (line 9), the algorithm updates the candidate set $\mathcal{D}^{t+1}$ to remove $i_{\min}$ and include $j$ (line 10). In this case, $z_{i_{\min}}^t$ must also be updated since $\mathcal{D}^{t+1}$ has been updated (line 11). If $|z_{i_{\min}}^t| < |z_j^t|$ does not hold (line 12), we set $z_j^t = 0$ because $j$ cannot be included in $\mathcal{D}^{t+1}$ (line 13).

For the outputs $\boldsymbol{z}^t$ and $\mathcal{D}^{t+1}$ of Algorithm 2, we have the following property:

**Lemma 4 (Consistency of outputs)** *For the outputs of Algorithm 2, $\boldsymbol{z}^t = \boldsymbol{\theta}^{t+1}$ and $\mathcal{D}^{t+1} = \mathrm{supp}(\boldsymbol{\theta}^{t+1})$ hold.*

The above lemma shows that Algorithm 2 returns the same $\boldsymbol{\theta}^{t+1}$ and $\mathrm{supp}(\boldsymbol{\theta}^{t+1})$ as those of the plain IHT while it prunes unnecessary computations given $z_{i_{\min}}^t, \overline{z}^t$, and $\mathcal{D}^t$. Specifically, we can use Algorithm 2 instead of lines 4–5 in Algorithm 1.

The computation cost of Algorithm 2 is as follows:

**Lemma 5 (Computation cost of updating candidate set)** *Let $u$ be the number of un-pruned computations at line 7 of Algorithm 2. If $\boldsymbol{G}$ and $\boldsymbol{X}^\top \boldsymbol{y}$ are precomputed, Algorithm 2 requires $\mathcal{O}(un)$ time and the worst time complexity is $\mathcal{O}(n^2)$ time.*

Lemma 5 shows that the cost of Algorithm 2 is small when the pruning rate is high because $u$ becomes small when the pruning rate is high. On the other hand, the worst time complexity of $\mathcal{O}(n^2)$ time is obtained with a low pruning rate. The asymptotic cost cannot be larger than that of the plain IHT

| **Algorithm 2** Update of candidate set | **Algorithm 3** Update of threshold |
|---|---|
| 1: **Input:** $z^t_{i_{\min}}, \overline{z}^t, \mathcal{D}^t$ | 1: **Input:** $z^t_{i_{\min}}, \underline{z}^t, \mathcal{D}^t$ |
| 2: **Output:** $z^t, \mathcal{D}^{t+1}$ | 2: **Output:** $z^{t'}_{i_{\min}}, z^{t'}, \mathcal{D}^{t'}$ |
| 3: $\mathcal{D}^{t+1} \leftarrow \mathcal{D}^t;$ | 3: $\mathcal{D}^{t'} \leftarrow \mathcal{D}^t$ |
| 4: **for** $j \notin \mathcal{D}^t$ **do** | 4: **for** $j \notin \mathcal{D}^t$ **do** |
| 5:    **if** $\|z^t_{i_{\min}}\| \geq \overline{z}^t_j$ **then** | 5:    **if** $\|z^t_{i_{\min}}\| < \underline{z}^t_j$ **then** |
| 6:       $z^t_j \leftarrow 0;$ | 6:       $\mathcal{D}^{t'} \leftarrow (\mathcal{D}^{t'} \setminus i_{\min}) \cup j;$ |
| 7:    **else** | 7:       compute $z^t_j;$ |
| 8:       compute $z^t_j;$ | 8:       $z^t_{i_{\min}} \leftarrow 0;$ |
| 9:       **if** $\|z^t_{i_{\min}}\| < \|z^t_j\|$ **then** | 9:       find $z^t_{i_{\min}}$ in $\mathcal{D}^{t'};$ |
| 10:          $\mathcal{D}^{t+1} \leftarrow (\mathcal{D}^{t+1} \setminus i_{\min}) \cup j;$ | 10: $z^{t'}_{i_{\min}} \leftarrow z^t_{i_{\min}};$ |
| 11:          $z^t_{i_{\min}} \leftarrow 0;$ find $z^t_{i_{\min}}$ in $\mathcal{D}^{t+1};$ | 11: $z^{t'} \leftarrow z^t;$ |
| 12:       **else** | |
| 13:          $z^t_j \leftarrow 0;$ | |

since the gradient computation of the plain IHT requires $\mathcal{O}(mn)$ or $\mathcal{O}(n^2)$ time. Nonetheless, the pruning rate needs to be increased to achieve higher speeds.

In Algorithm 2, $\|z^t_{i_{\min}}\|$ plays the role of the threshold for pruning at line 5. Therefore, we can raise the pruning rate for a larger threshold of $\|z^t_{i_{\min}}\|$ because $\|z^t_{i_{\min}}\| \geq \overline{z}^t_j$ at line 5 is easier to hold for a larger threshold. The next section introduces an efficient way to update $\|z^t_{i_{\min}}\|$ to a larger value before entering Algorithm 2.

### 4.3 Lower Bound and Update of Threshold

To update threshold $\|z^t_{i_{\min}}\|$ to a larger value, we utilize the lower bound $\underline{z}^t_j$ such that $\|z^t_j\| > \underline{z}^t_j$ holds for $j \notin \mathcal{D}^t$. $\underline{z}^t_j$ is defined as follows:

**Definition 3** *Let $t^*$ be $1 \leq t^* < t$ in Algorithm 1. Then, $\underline{z}^t_j$ at the $t$-th iteration in Algorithm 1 is computed as follows:*

$$\underline{z}^t_j = |G_j \theta^{t^*} + \eta(X^\top y)_j| - \|G_j\|_2 \|\theta^t - \theta^{t^*}\|_2, \tag{5}$$

*where $G = I - \eta X^\top X$.*

The following lemma shows that $\underline{z}^t_j$ is the lower bound of $\|z^t_j\|$:

**Lemma 6 (Lower bound)** *We have $\underline{z}^t_j \leq \|z^t_j\|$ when $\underline{z}^t_j$ is computed by Equation (5).*

Lemma 6 is derived from the reverse triangle inequality and the Cauchy–Schwarz inequality. Similarly to the computation cost of the upper bound, Equation (5) requires the following cost:

**Lemma 7 (Computation cost of lower bound)** *The computation cost of Equation (5) for all $j \in \{1, ..., n\}$ at the $t$-th iteration is $\mathcal{O}(n)$ time given $G$, $X^\top y$ and $|G_j \theta^{t^*} + \eta(X^\top y)_j|$.*

To update $\|z^t_{i_{\min}}\|$, we utilize the following lemma:

**Lemma 8 (Indices required for candidate set)** *Suppose that $\underline{z}^t$ is computed by using Equation (5), and the candidate set $\mathcal{D}^t$ is initialized as described in Definition 2 at the beginning of the iteration for $t > 1$. Let $z^t_{i_{\min}}$ be $z^t_i$ having the minimum $\|z^t_i\|$ in all $i \in \mathcal{D}^t$, and $i_{\min}$ be the index. Then, if $\|z^t_{i_{\min}}\| < \underline{z}^t_j$ holds for $j \notin \mathcal{D}^t$, $j$ is included in $\mathcal{D}^{t+1}$.*

The above lemma shows that we can identify indices included in $\mathcal{D}^{t+1}$ without computing the gradients by using the lower bound $\underline{z}^t$. The update procedure of $\mathcal{D}^t$ to $\mathcal{D}^{t+1}$ is described in

Algorithm 3. Since this $\mathcal{D}^{t+1}$ is used as the initial candidate set of Algorithm 2, we represent $\mathcal{D}^{t'}$ as $\mathcal{D}^{t+1}$ in Algorithm 3 to avoid confusion. The algorithm copies $\mathcal{D}^t$ to $\mathcal{D}^{t'}$ at line 3. Line 5 checks whether $|z_{i_{\min}}^t| < \underline{z}_j^t$ holds or not. If the equation holds, the algorithm updates $\mathcal{D}^{t'}$ to remove $i_{\min}$ and include $j$ (line 6). At this time, we also update $z_j^t$ and $z_{i_{\min}}^t$ to reflect the update of $\mathcal{D}^{t'}$ (lines 7–9). $z_{i_{\min}}^{t'}$ and $z^{t'}$ (lines 10–11) are used as $z_{i_{\min}}^t$ and $z^t$ in Algorithm 2.

The important point of Algorithm 3 is that the absolute value of the output $|z_{i_{\min}}^{t'}|$ is equal to or larger than the initial $|z_{i_{\min}}^t|$ as follows:

**Lemma 9 (Threshold increase)** *In Algorithm 3, $|z_{i_{\min}}^{t'}| \geq |z_{i_{\min}}^t|$ holds.*

From the above lemma, we can obtain large $|z_{i_{\min}}^{t'}|$ as the threshold $|z_{i_{\min}}^t|$ in Algorithm 2 by performing Algorithm 3 before entering Algorithm 2. As a result, we can expect Algorithm 2 to increase the pruning rate.

The computation cost of Algorithm 3 is as follows:

**Lemma 10 (Computation cost of threshold increase)** *Let $l$ be the number of indices that are determined to be included in $\mathcal{D}^{t'}$ at line 5 of Algorithm 3. If $\boldsymbol{G}$ and $\boldsymbol{X}^\top \boldsymbol{y}$ are precomputed, Algorithm 3 requires $\mathcal{O}(ln)$ time and the worst time complexity of Algorithm 3 is $\mathcal{O}(n^2)$ time.*

Similarly to Lemma 5 of Algorithm 2, the worst time complexity is not much more than that of IHT.

### 4.4  Algorithm

Algorithm 4 is the pseudocode of our method based on Algorithms 2 and 3. We first precompute $\boldsymbol{G}$ and $\boldsymbol{X}^\top \boldsymbol{y}$ for the upper and lower bounds (line 3). The main loop consists of two types of procedures (lines 4–22): the procedure for $t = t^*$ (lines 5–11) and the procedure for $t \neq t^*$ (lines 12–21). For the case of $t = t^*$, the algorithm updates the parameter vector the same as the plain IHT (lines 6–7) and computes the candidate set (line 8). We note that the computation result of line 6 is also used for computing the upper and lower bounds as shown in Definitions 1 and 3. Line 9 sets $t^*$ as $t$ and line 10 computes an interval $r \geq 1$ that determines which $t$ is the next $t^*$. Specifically, the next $t^*$ is determined as $t^* + r$. The computation way of $r$ is described later. $r'$ is the variable that monitors the interval. If $t = t^*$ does not hold (line 12), it computes $z_i^t$ for $i \in \mathcal{D}^t$ by using Equation (2) (line 13). Next, line 14 computes the lower bound by using Equation (5) on the basis of Lemma 6. Then, we find the initial threshold at line 15 and update it by using Algorithm 3 on the basis of Lemmas 8 and 9 (line 16). Then, we compute the upper bound by using Equation (4) on the basis of Lemma 1 (line 17). Line 18 computes $z^t$ by using Algorithm 2 while pruning unnecessary computations on the basis of Lemma 3. This $z^t$ can be handled as $\boldsymbol{\theta}^{t+1}$ by following Lemma 4 (line 19). We monitor the interval of the next $t^*$ at line 20 through $r'$. The algorithm repeats the above procedure until a stopping criterion is met (line 22). An example of a stopping criterion is relative tolerance [23].

**Automatic determination of $t^*$ via $r$.** In Algorithm 4, we need to compute interval $r$ at line 10 to determine $t^*$ because $\boldsymbol{\theta}^{t^*}$ appears in Equations (4) and (5) for computing the upper and lower bounds, respectively. Since the upper bound $\overline{z}_j^t$ and lower bound $\underline{z}_j^t$ are the approximation of $z_j^t$, we obtain the following error bound:

**Lemma 11** *Suppose that $\epsilon_j$ is computed as follows:*

$$\epsilon_j = 2\|\boldsymbol{G}_j\|_2 \|\boldsymbol{\theta}^t - \boldsymbol{\theta}^{t^*}\|_2. \tag{6}$$

*Then, $|\overline{z}_j^t - z_j^t| \leq \epsilon_j$ and $|\underline{z}_j^t - z_j^t| \leq \epsilon_j$ hold.*

From the above lemma, the magnitude of error bound $\epsilon_j$ depends on $t^*$ because of $\|\boldsymbol{\theta}^t - \boldsymbol{\theta}^{t^*}\|_2$ in Equation (6). If we set $r$ to a large value, the error bound can be large since $\|\boldsymbol{\theta}^t - \boldsymbol{\theta}^{t^*}\|_2$ tends to be large. In this case, the bounds are loose and it is difficult to hold $|z_{i_{\min}}^t| \geq \overline{z}_j^t$ in Lemma 3. As a result, the pruning rate will be low. On the other hand, if we set $r$ a small value, the algorithm frequently computes lines 5–11 although we can obtain tight bounds. Since line 6 requires $\mathcal{O}(n^2)$ time, the reduction of the computation cost will be small. To solve the above problem, we automatically determine $r$ on the basis of the current pruning rate that is defined as follows:

---
**Algorithm 4** Fast Iterative Hard Thresholding
---
1: **Input:** sparsity level $k$, step size $\eta$
2: **Initialization:** $\boldsymbol{\theta}^1 \leftarrow \mathbf{0}$, $t \leftarrow 1$, $t^* \leftarrow 1$, $r \leftarrow 0$, $r' \leftarrow 0$
3: computing $\boldsymbol{G}$ and $\boldsymbol{X}^\top \boldsymbol{y}$;       ▷ The precomputation for the upper and lower bounds
4: **repeat**                           ▷ The main loop
5:  **if** $r' = 0$ **then**       ▷ The precomputation for the upper and lower bounds
6:    $\boldsymbol{z}^t \leftarrow \boldsymbol{G}\boldsymbol{\theta}^t + \eta \boldsymbol{X}^\top \boldsymbol{y}$;      ▷ Computing $\boldsymbol{G}\boldsymbol{\theta}^t + \eta \boldsymbol{X}^\top \boldsymbol{y}$ used for $\underline{\boldsymbol{z}}^t$ and $\overline{\boldsymbol{z}}^t$
7:    $\boldsymbol{\theta}^{t+1} \leftarrow H_k(\boldsymbol{z}^t)$;               ▷ Updating the parameter
8:    $\mathcal{D}^t \leftarrow \mathrm{supp}(\boldsymbol{\theta}^{t+1})$;              ▷ Updating the candidate set
9:    $t^* \leftarrow t$;
10:    compute $r$ on the basis of automatic determination and $r' \leftarrow r$;
11:    $t \leftarrow t + 1$;
12:  **else**
13:    compute $\boldsymbol{z}_i^t$ for $i \in \mathcal{D}^t$ by Eqn. (2);
14:    compute $\underline{\boldsymbol{z}}^t$ by Eqn. (5);    ▷ Computing the lower bound on the basis of Lemma 6
15:    find $\boldsymbol{z}_{i_{\min}}^t$ in $\mathcal{D}^t$;
16:    update $\boldsymbol{z}_{i_{\min}}^t$, $\boldsymbol{z}^t$ and $\mathcal{D}^t$ by Algorithm 3;    ▷ Based on Lemmas 8 and 9
17:    compute $\overline{\boldsymbol{z}}^t$ by Eqn. (4);   ▷ Computing the upper bound on the basis of Lemma 1
18:    compute $\boldsymbol{z}^t$ and $\mathcal{D}^{t+1}$ by Algorithm 2;      ▷ Based on Lemma 3
19:    $\boldsymbol{\theta}^{t+1} \leftarrow \boldsymbol{z}^t$;       ▷ Updating the parameter on the basis of Lemma 4
20:    $r' \leftarrow r' - 1$;
21:    $t \leftarrow t + 1$;
22: **until** a stopping criterion is met
---

**Definition 4 (Pruning rate)** *Let $u_t$ be the number of un-pruned computations at line 7 in Algorithm 2 for the $t$-th iteration as defined in Lemma 5. Then, we define pruning rate $p_t$ at line 10 in Algorithm 4 for the $t$-th iteration as follows:*

$$p_t = \frac{n - k - u_{t-1}}{n - k} \times 100.0. \tag{7}$$

The unit of $p_t$ is percent. We compute $r$ at line 10 in Algorithm 4 on the basis of $p_t$ as follows:

$$r = \begin{cases} r + 1 & \text{if } p_t \geq 50.0 \\ \lceil r/2 \rceil & \text{if } p_t < 50.0. \end{cases} \tag{8}$$

$\lceil \cdot \rceil$ is the ceiling function. If the algorithm could prune half of the computations at the previous iteration, it increases interval $r$. If not, we update $r$ to $\lceil r/2 \rceil$ to reduce error bound $\epsilon_j$. Therefore, if the pruning rate is high, we can reduce the number of computations at line 6 by increasing the interval. If not, the interval becomes small and the upper and lower bounds are expected to be tight. This rule performs well in our experiments.

### 4.5 Analysis

The computation cost of Algorithm 4 is as follows:

**Theorem 1 (Computation cost)** *Let $u'$ and $l'$ be the total numbers of $u$ and $l$ in Lemmas 5 and 10 in Algorithm 4, respectively. Suppose that $r$ is a constant for simplicity, $\tau$ is the total number of main loops in Algorithm 4, and $\tau'$ is the number for which line 12 holds. Then, the computation cost of Algorithm 4 is $\mathcal{O}(n^2(m + \frac{\tau}{r+1}) + n(l' + u' + \tau' k))$ time, and the worst time complexity is $\mathcal{O}(n^2(m + \tau))$ time.*

The plain IHT of Algorithm 1 requires $\mathcal{O}(n^2(m + \tau))$ time if $\boldsymbol{G}$ and $\boldsymbol{X}^\top \boldsymbol{y}$ are precomputed. Therefore, the worst time complexity of our method is the same as the cost of the plain IHT.

For the optimization result, we obtain the following theorem:

**Theorem 2 (Optimization result)** *Suppose that Algorithm 4 has the same hyperparameters as those of the plain IHT of Algorithm 1. Then, Algorithm 4 yields the same parameter vector and objective value as Algorithm 1.*

Theorem 2 guarantees accuracy of our method. Our method prunes unnecessary computations without degrading accuracy.

For the upper and lower bounds, the following property holds:

**Theorem 3 (Convergence of upper and lower bounds)** *Suppose that Algorithm 4 converges as* $\boldsymbol{\theta}^t = \boldsymbol{\theta}^{t^*}$. *Then, we obtain* $\epsilon_j = 0$ *for* $j \in \{1, ..., n\}$ *where* $\epsilon_j$ *is the error bound of the upper and lower bounds defined in Lemma 11.*

Theorem 3 shows that the upper bound $\overline{z}_j^t$ and lower bound $\underline{z}_j^t$ become the exact value of $z_j^t$ when $\boldsymbol{\theta}^t = \boldsymbol{\theta}^{t^*}$ holds. Therefore, our method accurately prunes unnecessary computations when the condition is satisfied.

# 5  Experiment

We evaluated the processing time and accuracy of our method on feature selection tasks. We performed experiments on five datasets from the LIBSVM [9] and OpenML [34]: *gisette*, *robert*, *ledgar*, *real-sim*, and *epsilon*. The sizes of the input matrices are $6000 \times 5000$, $10000 \times 7200$, $60000 \times 19996$, $72309 \times 20958$, and $400000 \times 2000$, respectively. We evaluated the processing time and accuracy on $k = \{1, 5, 10, 20, 40, 80, 160, 320, 640, 1280\}$. We compared our method with the plain IHT (IHT), Regularized IHT (RegIHT) [2], and Accelerated IHT (AccIHT) [23]. RegIHT is the fastest method among the methods using an adaptive regularization technique [1]. Since RegIHT has the hyperparameter of weight step size $c$, we tried the setting of $c = \{k, k/10, k/100\}$ on the basis of the original paper. AccIHT improves the convergence of IHT by utilizing the momentum. We tried $\mu = \{0.025, 0.25, 2.5\}$ for the momentum step size $\mu$ where the value of $0.25$ is the one recommended in the original paper. We set step sizes of all the methods $\eta = 1/\lambda$ where $\lambda$ is the largest eigen value of $\boldsymbol{X}^\top \boldsymbol{X}$ by following [23]. We stopped these methods when the relative tolerance of the parameter vector dropped below $10^{-5}$ [23, 15]. All the experiments were conducted on a 3.20 GHz Intel CPU with six cores and 64 GB of main memory.

## 5.1  Processing Time

Figure 1 (a)–(e) compare the processing times on logarithmic scale. Our method was up to 73 times faster than IHT and outperformed all the baselines in all the settings. Because our method is based on the pruning, it achieved a large speedup factor for smaller $k$. Even when $k$ increased, the processing time was significantly shorter than the baselines.

We note that our method does not need hyperparameter tuning due to using the automatic determination technique described in Section 4.4. Specifically, practitioners only need to specify $k$ to use our method the same as the plain IHT. On the other hand, the baselines of RegIHT and AccIHT require additional hyperparameter tuning for the weight step size and the momentum step size, respectively.

**Number of Gradient Computations.** Figure 1 (f) compares the number of gradient computations between our method and the plain IHT on the gisette dataset. Our method reduced the number of computations by up to $98.87\%$. The result shows the effectiveness of our pruning strategy and supports the reduction in processing times of Figure 1 (a)–(e).

**Processing Time with Large Step Size.** Since the error bound of upper and lower bounds depends on $\|\boldsymbol{\theta}^t - \boldsymbol{\theta}^{t^*}\|_2$ in Equation (6), a large step size may incur a potential decrease in the pruning rate and our method may be slow down. To address this concern, we conducted an experiment to evaluate the processing times with an increased step size. We increased the step size to 10 times larger than that used in Figure 1 (a) on the gisette dataset. Figure 1 (g) shows the results, exhibit a similar trend to Figure 1 (a). Our method was able to speed up IHT even with the larger step size. This success is due to our automatic determination of $t^*$, which adjusts the pruning rate during optimization, as described in Definition 4 and Equation (8). In contrast, AccIHT failed to converge in some cases due to the momentum, preventing us from evaluating the processing times for those instances. While

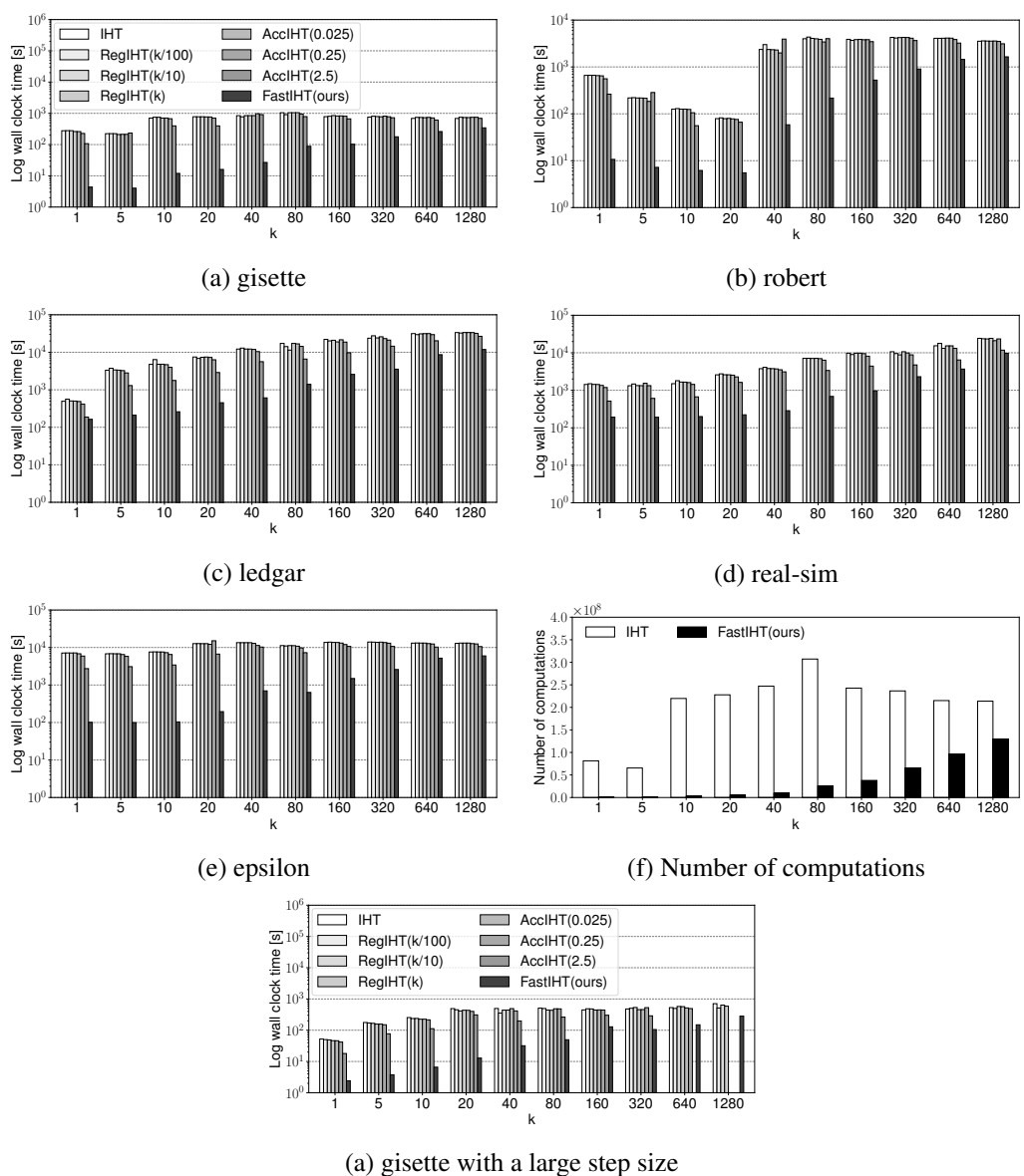

(a) gisette        (b) robert

(c) ledgar        (d) real-sim

(e) epsilon        (f) Number of computations

(a) gisette with a large step size

Figure 1: (a)–(e): Comparisons of log processing times for each dataset and $k$. (f): Comparison of number of gradient computations on gisette. (g): Comparisons of log processing times for gisette with a large step size. Some results of AccIHT are omitted since they could not converge.

AccIHT reduces the number of iterations by using the momentum, our method reduces the cost per iteration on the basis of the pruning strategy. This result shows an advantage of the pruning approach.

## 5.2 Accuracy

Theorem 2 guarantees that our method achieves the same results as the plain IHT. To verify the theorem, we compared the objective values between our method and the plain IHT. Table 1 shows the results for $k = \{1, 20, 160, 1280\}$, and our method achieved the same objective values as the plain IHT. We obtained the same trend results as above in the other settings of $k$. We note that our method also yielded the same support of nonzero elements in the parameter vector and their coefficients as the plain IHT though the results are omitted. These results support our theoretical result for Theorem 2. In addition, our method also ensures that the parameter vector of each iteration matches perfectly

Table 1: Objective values of the plain IHT and our method for $k = \{1, 20, 160, 1280\}$.

| dataset | method | $k = 1$ | $k = 20$ | $k = 160$ | $k = 1280$ |
|---|---|---|---|---|---|
| gisette | IHT | $56.01 \times 10^{-2}$ | $31.99 \times 10^{-2}$ | $14.01 \times 10^{-2}$ | $80.73 \times 10^{-3}$ |
| | ours | $56.01 \times 10^{-2}$ | $31.99 \times 10^{-2}$ | $14.01 \times 10^{-2}$ | $80.73 \times 10^{-3}$ |
| robert | IHT | $99.03 \times 10^{-1}$ | $91.23 \times 10^{-1}$ | $73.56 \times 10^{-1}$ | $66.24 \times 10^{-1}$ |
| | ours | $99.03 \times 10^{-1}$ | $91.23 \times 10^{-1}$ | $73.56 \times 10^{-1}$ | $66.24 \times 10^{-1}$ |
| ledgar | IHT | $12.76 \times 10^{2}$ | $82.48 \times 10^{1}$ | $50.70 \times 10^{1}$ | $35.35 \times 10^{1}$ |
| | ours | $12.76 \times 10^{2}$ | $82.48 \times 10^{1}$ | $50.70 \times 10^{1}$ | $35.35 \times 10^{1}$ |
| real-sim | IHT | $86.47 \times 10^{-2}$ | $63.84 \times 10^{-2}$ | $40.32 \times 10^{-2}$ | $23.16 \times 10^{-2}$ |
| | ours | $86.47 \times 10^{-2}$ | $63.84 \times 10^{-2}$ | $40.32 \times 10^{-2}$ | $23.16 \times 10^{-2}$ |
| epsilon | IHT | $93.50 \times 10^{-2}$ | $67.24 \times 10^{-2}$ | $44.93 \times 10^{-2}$ | $43.03 \times 10^{-2}$ |
| | ours | $93.50 \times 10^{-2}$ | $67.24 \times 10^{-2}$ | $44.93 \times 10^{-2}$ | $43.03 \times 10^{-2}$ |

with that of the plain IHT from Lemma 4. This property is not obtained in previous acceleration methods based on the momentum such as AccIHT.

## 6 Conclusion

We accelerated iterative hard thresholding (IHT) by safely pruning unnecessary gradient computations. The main idea is to efficiently maintain the candidate set, which contains indices of nonzero elements in the parameter vector, during optimization. Before computing the gradients for each iteration, we prune unnecessary elements for the candidate set by utilizing the upper bound. To raise the pruning rate, we update the threshold to determine whether an element is included or not in the candidate set by using the lower bound. Our method guarantees the same optimization results as the plain IHT because our pruning is safe. In addition, it does not need additional hyperparameter tuning. Experiments show that our method is up to 73 times faster than IHT without degrading accuracy. As future work, we plan to extend our method to general convex loss functions with sparsity-inducing norms to enhance more applications.

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

# Appendix

## A  Proofs

### A.1  Lemma 1

**Proof**  *From Equation (2), we obtain the following equation:*

$$
\begin{aligned}
\boldsymbol{z}^t &= \boldsymbol{z}^t + \boldsymbol{z}^{t^*} - \boldsymbol{z}^{t^*} \\
&= (\boldsymbol{I} - \eta \boldsymbol{X}^\top \boldsymbol{X})\boldsymbol{\theta}^t + \eta \boldsymbol{X}^\top \boldsymbol{y} + (\boldsymbol{I} - \eta \boldsymbol{X}^\top \boldsymbol{X})\boldsymbol{\theta}^{t^*} + \eta \boldsymbol{X}^\top \boldsymbol{y} - (\boldsymbol{I} - \eta \boldsymbol{X}^\top \boldsymbol{X})\boldsymbol{\theta}^{t^*} - \eta \boldsymbol{X}^\top \boldsymbol{y} \\
&= \boldsymbol{G}\boldsymbol{\theta}^{t^*} + \eta \boldsymbol{X}^\top \boldsymbol{y} + \boldsymbol{G}(\boldsymbol{\theta}^t - \boldsymbol{\theta}^{t^*}).
\end{aligned}
$$

*By using the triangle inequality and the Cauchy–Schwarz inequality, we have the following inequality from the above equation:*

$$
\begin{aligned}
|\boldsymbol{z}_j^t| &\leq |\boldsymbol{G}_j \boldsymbol{\theta}^{t^*} + \eta (\boldsymbol{X}^\top \boldsymbol{y})_j| + |\boldsymbol{G}_j(\boldsymbol{\theta}^t - \boldsymbol{\theta}^{t^*})| \\
&\leq |\boldsymbol{G}_j \boldsymbol{\theta}^{t^*} + \eta (\boldsymbol{X}^\top \boldsymbol{y})_j| + \|\boldsymbol{G}_j\|_2 \|\boldsymbol{\theta}^t - \boldsymbol{\theta}^{t^*}\|_2,
\end{aligned}
$$

*which completes the proof.*  □

### A.2  Lemma 2

**Proof**  *Computing $\|\boldsymbol{\theta}^t - \boldsymbol{\theta}^{t^*}\|_2$ in Equation (4) requires $\mathcal{O}(n)$ time. Therefore, the cost of Equation (4) at the $t$-th iteration is $\mathcal{O}(n)$ time if $\boldsymbol{G}$, $\boldsymbol{X}^\top \boldsymbol{y}$ and $|\boldsymbol{G}_j \boldsymbol{\theta}^{t^*} + \eta (\boldsymbol{X}^\top \boldsymbol{y})_j|$ are precomputed.*  □

### A.3  Lemma 3

**Proof**  *If $|\boldsymbol{z}_{i_{\min}}^t| \geq \overline{\boldsymbol{z}}_j^t$ holds for $j \notin \mathcal{D}^t$, we obtain $|\boldsymbol{z}_{i_{\min}}^t| \geq |\boldsymbol{z}_j^t|$ from $\overline{\boldsymbol{z}}_j^t \geq |\boldsymbol{z}_j^t|$. In this case, since $|\boldsymbol{z}_{i_{\min}}^t|$ is the minimum $|\boldsymbol{z}_i^t|$ for all $i \in \mathcal{D}^t$, $|\boldsymbol{z}_j^t|$ cannot be larger than $|\boldsymbol{z}_i^t|$ for all $i \in \mathcal{D}^t$. Therefore, $j$ cannot be included in $\mathcal{D}^{t+1}$.*  □

### A.4  Lemma 4

**Proof**  *In Algorithm 2, if $|\boldsymbol{z}_{i_{\min}}^t| \geq \overline{\boldsymbol{z}}_j^t$ holds for $j \notin \mathcal{D}^t$, $j$ cannot be included in $\mathcal{D}^{t+1}$ by Lemma 3 (line 5). In addition, $|\boldsymbol{z}_{i_{\min}}^t|$ does not decrease compared to the previous $|\boldsymbol{z}_{i_{\min}}^t|$ when it is updated at line 11. Therefore, $|\boldsymbol{z}_{i_{\min}}^t| \geq \overline{\boldsymbol{z}}_j^t$ always holds even after $|\boldsymbol{z}_{i_{\min}}^t|$ is updated. As a result, since $|\boldsymbol{z}_j^t|$ cannot be included in the $k$-largest elements in the magnitude of $\boldsymbol{z}^t$, $j$ cannot be included in $\mathrm{supp}(\boldsymbol{\theta}^{t+1})$. If $|\boldsymbol{z}_{i_{\min}}^t| < \overline{\boldsymbol{z}}_j^t$ holds (line 7), Algorithm 2 exactly computes $\boldsymbol{z}_j^t$ (line 8) and checks whether $|\boldsymbol{z}_j^t|$ is included or not in the $k$-largest elements in the magnitude of $\boldsymbol{z}^t$ (lines 9–13). Therefore, this part exactly computes $\boldsymbol{\theta}^{t+1} = H_k(\boldsymbol{z}^t)$ for $j$. Since $j$ cannot be included in $\mathrm{supp}(\boldsymbol{\theta}^{t+1})$ for the case of $|\boldsymbol{z}_{i_{\min}}^t| \geq \overline{\boldsymbol{z}}_j^t$ and the other case exactly computes $\boldsymbol{\theta}^{t+1} = H_k(\boldsymbol{z}^t)$, $\boldsymbol{z}^t$ matches $\boldsymbol{\theta}^{t+1}$. In addition, since $\mathcal{D}^{t+1} = \mathrm{supp}(\boldsymbol{z}^t)$ holds from line 10, $\mathcal{D}^{t+1}$ matches $\mathrm{supp}(\boldsymbol{\theta}^{t+1})$.*  □

### A.5  Lemma 5

**Proof**  *For un-pruned $j$ at line 7 in Algorithm 2, computing $\boldsymbol{z}_j^t$ (line 8) requires $\mathcal{O}(n)$ time if $\boldsymbol{G}$ and $\boldsymbol{X}^\top \boldsymbol{y}$ are precomputed. When $|\boldsymbol{z}_{i_{\min}}^t| < |\boldsymbol{z}_j^t|$ holds (line 9), we need to find $\boldsymbol{z}_{i_{\min}}^t$ in $\mathcal{D}^{t+1}$ in addition to the above computation. The cost is $\mathcal{O}(k)$ time because of $|\mathcal{D}^t| = k$. Because the number of un-pruned computations is $u$, the total cost of Algorithm 2 is $\mathcal{O}(u(n + k))$ time. At this time, since $k \ll n$, the final cost is $\mathcal{O}(un)$ time.*

*If Algorithm 2 cannot prune any computation for $j \notin \mathcal{D}^t$, the number of un-pruned computations $u$ equals $n - k$. Since $k \ll n$, the worst time complexity is $\mathcal{O}(n^2)$ time.*  □

## A.6 Lemma 6

**Proof** *Similarly to the proof of Lemma 1, we obtain the following inequality by using Equation (2), the reverse triangle inequality, and the Cauchy–Schwarz inequality:*

$$|\boldsymbol{z}_j^t| \geq |\boldsymbol{G}_j \boldsymbol{\theta}^{t^*} + \eta(\boldsymbol{X}^\top \boldsymbol{y})_j| - |\boldsymbol{G}_j(\boldsymbol{\theta}^t - \boldsymbol{\theta}^{t^*})|$$
$$\geq |\boldsymbol{G}_j \boldsymbol{\theta}^{t^*} + \eta(\boldsymbol{X}^\top \boldsymbol{y})_j| - \|\boldsymbol{G}_j\|_2 \|\boldsymbol{\theta}^t - \boldsymbol{\theta}^{t^*}\|_2,$$

*which completes the proof.* □

## A.7 Lemma 7

**Proof** *Similarly to the proof of Lemma 2, computing $\|\boldsymbol{\theta}^t - \boldsymbol{\theta}^{t^*}\|_2$ in Equation (5) requires $\mathcal{O}(n)$ time. Therefore, the cost of Equation (5) at the $t$-th iteration is $\mathcal{O}(n)$ time if $\boldsymbol{G}$, $\boldsymbol{X}^\top \boldsymbol{y}$ and $|\boldsymbol{G}_j \boldsymbol{\theta}^{t^*} + \eta(\boldsymbol{X}^\top \boldsymbol{y})_j|$ are precomputed.* □

## A.8 Lemma 8

**Proof** *When $|\boldsymbol{z}_{i_{\min}}^t| < \underline{\boldsymbol{z}}_j^t$ holds, $|\boldsymbol{z}_{i_{\min}}^t| < |\boldsymbol{z}_j^t|$ holds because we have $\underline{\boldsymbol{z}}_j^t < |\boldsymbol{z}_j^t|$. Since $|\boldsymbol{z}_j^t|$ is larger than $|\boldsymbol{z}_{i_{\min}}^t|$, which is the minimum $|\boldsymbol{z}_i^t|$ for $i \in \mathcal{D}^t$, $j$ is included in $\mathcal{D}^{t+1}$.* □

## A.9 Lemma 9

**Proof** *When $|\boldsymbol{z}_{i_{\min}}^t| < \underline{\boldsymbol{z}}_j^t$ holds at line 5 in Algorithm 3, the corresponding $|\boldsymbol{z}_j^t|$ is larger than $|\boldsymbol{z}_{i_{\min}}^t|$ since we have $\underline{\boldsymbol{z}}_j^t < |\boldsymbol{z}_j^t|$. In addition, $\boldsymbol{z}_{i_{\min}}^t$ is set to zero at line 8. Thus, the absolute value of the new $\boldsymbol{z}_{i_{\min}}^t$ found at line 9 is equal to or larger than the old one. If $|\boldsymbol{z}_{i_{\min}}^t|$ is not updated in Algorithm 3, $|\boldsymbol{z}_{i_{\min}}^{t'}|$ is equal to the initial $|\boldsymbol{z}_{i_{\min}}^t|$. Therefore, $|\boldsymbol{z}_{i_{\min}}^{t'}| \geq |\boldsymbol{z}_{i_{\min}}^t|$ holds.* □

## A.10 Lemma 10

**Proof** *Similarly to the proof of Lemma 5, computing $\boldsymbol{z}_j^t$ (line 7) requires $\mathcal{O}(n)$ time in Algorithm 3 if $\boldsymbol{G}$ and $\boldsymbol{X}^\top \boldsymbol{y}$ are precomputed. In addition, we need to find $\boldsymbol{z}_{i_{\min}}^t$ in $\mathcal{D}^{t'}$ at $\mathcal{O}(k)$ time (line 9). Therefore, the total cost of Algorithm 3 is $\mathcal{O}(l(n+k))$ time. At this time, since $k \ll n$, the final cost is $\mathcal{O}(ln)$ time.*

*If all $j \notin \mathcal{D}^t$ are determined to be included in $\mathcal{D}^{t'}$ at line 5, its number is $l = n - k$. Since $k \ll n$, the worst time complexity is $\mathcal{O}(n^2)$ time.* □

## A.11 Lemma 11

**Proof** *From Equations (4) and (5), $|\overline{\boldsymbol{z}}_j^t - \underline{\boldsymbol{z}}_j^t| = 2\|\boldsymbol{G}_j\|_2 \|\boldsymbol{\theta}^t - \boldsymbol{\theta}^{t^*}\|_2 = \epsilon_j$. Then, for the upper bound, $|\overline{\boldsymbol{z}}_j^t - \boldsymbol{z}_j^t| \leq |\overline{\boldsymbol{z}}_j^t - \underline{\boldsymbol{z}}_j^t| = \epsilon_j$ holds. Similarly to the upper bound, we also obtain the inequality in the lemma for the lower bound.* □

## A.12 Theorem 1

**Proof** *For line 3 in Algorithm 4, computing $\boldsymbol{G}$ and $\boldsymbol{X}^\top \boldsymbol{y}$ require $\mathcal{O}(n^2 m)$ and $\mathcal{O}(nm)$ times, respectively. Next, lines 6 and 7 require $\mathcal{O}(n^2)$ and $\mathcal{O}(n \log k)$ times, respectively. Since lines 6–11 are performed $\tau/(r+1)$ times, the cost of this part is $\mathcal{O}(n^2\tau/(r+1))$ time. For lines 13–21, the costs of $\mathcal{O}(nk)$ (line 13), $\mathcal{O}(n)$ (line 14), $\mathcal{O}(k)$ (line 15), $\mathcal{O}(ln)$ (line 16), $\mathcal{O}(n)$ (line 17), and $\mathcal{O}(un)$ (line 18) are required. Since this part is performed $\tau'$ times, the cost represented as $\mathcal{O}(n(l' + u' + \tau'k))$ time by using $l'$ and $u'$. Therefore, the total cost of Algorithm 4 is $\mathcal{O}(n^2(m + \frac{\tau}{r+1}) + n(l' + u' + \tau'k))$ time.*

*The case of $l = u = n$ and $k = n$ yields the worst time complexity. In this case, the cost of lines 13–21 is $\mathcal{O}(\tau'n^2)$ time. In addition, the cost of lines 6–11 can be represented as $\mathcal{O}((\tau - \tau')n^2)$ time. As a result, the cost is $\mathcal{O}(n^2(m + \tau))$ time.* □

### A.13 Theorem 2

**Proof** *For lines 6–7 in Algorithm 4, line 7 returns the same $\boldsymbol{\theta}^{t+1}$ as line 5 in Algorithm 1 since both procedures are the same. Line 19 in Algorithm 4 also returns the same $\boldsymbol{\theta}^{t+1}$ as line 5 in Algorithm 1 because of Lemma 4. Thus, the sequence of $\boldsymbol{\theta}^{t+1}$ in Algorithm 4 is the same as that of Algorithm 1 if both algorithms have the same hyperparameters. Therefore, Algorithm 4 returns the same parameter vector and objective value as Algorithm 1.* □

### A.14 Theorem 3

**Proof** *If $\boldsymbol{\theta}^t = \boldsymbol{\theta}^{t^*}$ holds, $\|\boldsymbol{\theta}^t - \boldsymbol{\theta}^{t^*}\|_2 = 0$ holds in Equation (6). As a result, we obtain $\epsilon_j = 0$ in Lemma 11.* □

## B Computation Cost of IHT with Sparse Matrices

If $\boldsymbol{G}$ and $\boldsymbol{X}^\top \boldsymbol{y}$ are precomputed, the cost of the gradient computations in IHT is $\mathcal{O}(n^2)$ time due to the third equation of Equation (2). If we use a sparse matrix for the parameter vector, the cost is $\mathcal{O}(nk)$ time. However, since the parameter vector changes for each iteration and the sparse matrix must be re-constructed for each iteration, it is rarely used for the parameter vector in practice due to the overhead of the sparse matrix construction[3]. We note that the sparse matrices may be used for the design matrices whose elements do not change for each iteration.

## C Broader Impacts

This paper presents work whose goal is to advance the field of machine learning. There are many potential societal consequences of our work, none which we feel must be specifically highlighted here.

---

[3]`https://scikit-learn.org/stable/modules/generated/sklearn.linear_model.Lasso.html`

