# OpenReview forum: "Fast Iterative Hard Thresholding Methods with Pruning Gradient Computations"
_NeurIPS.cc/2024/Conference — NeurIPS 2024 poster_

### Official Review · Reviewer_FTQw · 2024-07-04

**Soundness:** 3
**Presentation:** 3
**Contribution:** 3
**Rating:** 7
**Confidence:** 3

**Summary:**

Iterative hard thresholding (IHT) is used to select the k most important features in an ordinary least squares (OLS) linear regression model, that is, the model parameter vector is constrained to have only k non-zero entries. It seems that most practical IHT methods to solve the constrained problem are based on gradient descent. The running time of iterative methods until convergence depends on (a) the computational cost of each iteration, and (b) the number of iterations. If we understand it correctly, previous work has focussed on reducing the number of iterations by using some form of regularization (improving the smoothness of the problem) or by exploiting information from previous iterations, such as, momentum. The paper under review seems first to propose a method for reducing the computation cost of the iterations by avoiding the computation of unnecessary entries in the gradient. This is achieved by maintaining upper and lower bounds for each entry of the full OLS parameter vector at each iteration. The bounds can be used to prune computations of unnecessary entries in the gradient. The method is guaranteed to give the same results as plain IHT.

**Strengths:**

The problem is well stated and relevant. The proposed idea to reduce the cost per iteration is, as far as I can tell, novel. Its realization is technically non-trivial, but well described. I have not checked the proofs in the appendix, but intuitively the results make sense.  In general, the paper is well written. Actually, I enjoyed reading it.

**Weaknesses:**

I do not see any major weakness.

Maybe the presentation can be improved in some parts. Here are some suggestions:

Line 51: m-dimensionAL vector
Line 63:  ... if it uses a heap ...
Line 137: The ASYMPTOTIC cost ... [in some situations the actual cost should be higher]
Lines 239 and 241: Avoid starting a sentence with references.
Table 1: The information content is really low.

**Questions:**

1. Would it be possible to combine your technique with techniques that aim to reduce the number of iterations? For instance, the techniques described in references 4 and 26, or 8 and 20?

2. Why did you not include the methods from references 4, 26, 8, and 20 in your experiments? Are they superseded by references 2 and 19?

**Limitations:**

The scope of the problem addressed by the paper is clearly stated. Within this scope, I do not see limitations.

---

> ### Author Rebuttal · Authors · 2024-08-06
>
> We sincerely appreciate the reviewer’s positive feedback on our paper. As mentioned in the reviewer's comment, previous methods for accelerating IHT have focused on reducing the number of iterations. In contrast, our approach accelerates IHT by reducing the computation cost per iteration. Accelerating IHT is challenging due to its non-convexity, resulting from $\ell_0$ cardinality constraints in Problem (1), and there are relatively few acceleration methods for IHT compared to those for $\ell_1$ regularization, such as lasso. Therefore, we hope that the proposed method will offer a new direction for speeding up IHT.
>
> We also thank the reviewer for the valuable suggestions on improving the presentation. We will revise the paper accordingly.
>
> Below are our responses to the questions:
>
> > Would it be possible to combine your technique with techniques that aim to reduce the number of iterations? For instance, the techniques described in references 4 and 26, or 8 and 20?
>
> Yes, it is relatively straightforward to combine these methods with our proposed method. This is because these methods utilize a linear combination of parameters to reduce the number of iterations. For example, the acceleration methods in [8] and [20] can be integrated with our method as follows:
>
> 1. Compute the linear combination of parameters at the $t$-th iteration $\phi^{t}$, by following Equation (21) in [8] or Equation (19) in [20].
>
> 2. Set $\theta^{t}=\phi ^{t}$. Then, we can compute the upper and lower bounds using Equations (4) and (5) in our paper, respectively.
>
> 3. Perform Algorithm 4 in our paper based on these upper and lower bounds.
>
> We are currently working on combining these acceleration methods with our approach and appreciate the valuable feedback.
>
> > Why did you not include the methods from references 4, 26, 8, and 20 in your experiments? Are they superseded by references 2 and 19?
>
> We sincerely thank the reviewer for the insightful comment. References [4, 26] utilize acceleration techniques known as double over-relaxation. However, the theoretical guarantees for these methods are limited. Specifically, the method in [26] lacks theoretical guarantee, and the method in [4] is guaranteed to converge only when the acceleration technique is applied under conditions where the objective function decreases. In contrast, [8, 20] provide convergence proofs when using Nesterov’s acceleration [24] and squared loss. Additionally, [19] examines the properties of the momentum step in Nesterov’s acceleration. Since the algorithms in [8, 20, 19] are almost the same, we consider [8, 20] to be superseded by [19] in our paper.
>
> Unlike the above methods, [2] adds squared $\ell_2$ regularization to Problem (1), and is completely different from the above papers. Therefore, we have added it to the baseline in our experiment.
>
> We would like to incorporate the above explanation into the related work section.

---

> > ### Comment · Reviewer_FTQw · 2024-08-09
> >
> > Thank you for answering my questions. As you have noticed, I like your paper. To make a clear statement, I will increase my score to accept.

---

> > > ### Author Response · Authors · 2024-08-09
> > >
> > > We are pleased to receive such a positive response and sincerely thank the reviewer for raising the score. If the reviewer has any further questions or suggestions, please do not hesitate to contact us.

---

### Official Review · Reviewer_abvp · 2024-07-05

**Soundness:** 1
**Presentation:** 1
**Contribution:** 2
**Rating:** 5
**Confidence:** 4

**Summary:**

This paper studies iterative hard thresholding (IHT) as a canonical method for sparse linear regression. With precomputed X^TX and X^Ty, the computational cost of the algorithm is dominated by the gradient updates. To reduce the computational cost, this work proposed a pruning procedure at each step of IHT to only compute certain elements of the gradient vector.

**Strengths:**

Reducing the computational cost of IHT is an interesting and important problem.
Numerical results indicate a significant reduction in running time by employing the pruning procedure proposed in the paper, without compromising the estimation accuracy.

**Weaknesses:**

Major:
* It is unclear to me why the pruning strategy is defined the way it is.
* Consider adding more literature review: it is not clear how big of a gap exists in the literature that this work is trying to bridge. Section 4 Related Work feels out of place. Could consider moving this to the beginning of the paper.
* A general suggestion: consider adding more explanation on the rationale behind each technical definition/ result and why it is defined/ stated in that particular way. For instance, it may be helpful to mention that Lemma 1 is derived from the triangle inequality + Holders inequality.
* I think some definitions are stated in the form of lemmas which they should not be, e.g. Lemmas 3 and 8. In general, I think some lemmas are unnecessary or can be combined.
* Presentation is a bit long-winded at places, e.g. third paragraph

Minor:
* “73 times”: a more precise statement may be more useful; briefly describe the dimensions of the problem etc
* line 60: it is more consistent to use “problem in (1)” instead of “Problem 1”
* Notation:
    * I find it more natural to use boldface X as the design matrix
    * it is a bit confusing to denote the hard thresholding operator by Pk, hk is more common
    * \mathbb{D}^t and \mathbb{I} are unconventional notations for sets. Use \mathcal{D}^t and \mathcal{I} instead.

**Questions:**

* I find the terminology "pruning" a bit confusing. My understanding is that it refers to setting a particular entry  $z_j^t$ to zero depending on whether or not $\bar{z}_{j}^t $ is less than     a certain threshold. Is this correct? If yes, then maybe "thresholding" is a better terminology?
* By "pruning is safe", do you mean that the pruning step does not compromise the accuracy of IHT at all? Can you prove this more explicitly?
* I think X^TX takes O(n^2m) computational cost instead of O(mn)?
* Did you mean to use f(\theta) instead of 1/2||y-X\theta||_2^2 in (1)?

**Limitations:**

I'd suggest the author add one or two sentences in the final Section to discuss the limitations of this work.

---

> ### Author Rebuttal · Authors · 2024-08-06
>
> We sincerely appreciate the reviewer's thoughtful review of our paper. Below are our answers to the questions and weaknesses raised by the reviewer.
>
> > I find the terminology "pruning" a bit confusing. My understanding is that it refers to setting a particular entry $z_{j}^{t}$ to zero depending on whether or not $\bar{z}^{t}_{j}$ is less than a certain threshold. Is this correct? If yes, then maybe "thresholding" is a better terminology?
>
> A1. Yes, that is correct. We chose the term "pruning" instead of "thresholding" to distinguish our procedure from the thresholding procedure of the hard thresholding operator $P_{k}$​. The thresholding of $P_{k}$ sets unnecessary $z_{j}^{t}$ to zero **after** computing the gradient and updating all the parameters. In contrast, our method's pruning sets unnecessary  $z_{j}^{t}$ to zero **before** computing the gradient and updating the parameters.
>
> > By "pruning is safe", do you mean that the pruning step does not compromise the accuracy of IHT at all? Can you prove this more explicitly?
>
> A2. Yes, our pruning method is theoretically guaranteed to maintain the same accuracy as the original IHT. This is demonstrated in Theorem 2, with detailed proof provided in the appendix. Below, we outline the proof:
>
> 1. The original IHT selects the top-$k$ parameters in absolute values. In contrast, our method uses a candidate set $\mathbb{D}^{t}$ with cardinality $k$, as described in Definition 2, and updates $\mathbb{D}^{t}$ to include the top-$k$ parameters during optimization.
>
> 2. At the beginning of each iteration, our method initializes $\mathbb{D}^{t}$ to include the indices of the current top-$k$ parameters (line 103). For the parameters corresponding to $\mathbb{D}^{t}$, we compute their gradients to obtain the exact $z_{i}^{t}$ for $i$ in $\mathbb{D}^{t}$  (line 13 in Algorithm 4). We then set $z_{i_{\rm min}}^{t}$ as the minimum $z_{i}^{t}$​ in $\mathbb{D}^{t}$, as shown in Lemma 3.
>
> 3. Next, we consider the upper bound of $z_{j}^{t}$​ for parameters not included in $\mathbb{D}^{t}$. This upper bound  $\bar{z}^{t}\_{j}$  is given by Definition 1 and Lemma 1. If  $\bar{z}^{t}\_{j} \leq z\_{i\_{\rm min}}^{t}$ holds,​ $z_{j}^{t}\leq z\_{i\_{\rm min}}^{t}$ also holds.​ Therefore, the $j$-th parameter must not be included in $\mathbb{D}^{t}$. In this case, the $j$-th parameter​ can be safely pruned without computing the gradient (Lemma 3). For parameters that do not satisfy the condition, our method exactly computes $z_{j}^{t}$​ as shown in line 8 of Algorithm 2. Consequently,  $\mathbb{D}^{t}$ is exactly the same as the top-$k$ parameters of the original IHT (Lemma 4).
>
> As described above, since our method prunes parameters that must be zero, pruning the corresponding gradient computations does not affect the final top-$k$ parameters. Additionally, our method not only achieves the same final accuracy as the original IHT but also ensures that the parameters during optimization match perfectly with those of the original IHT. We would like to highlight that this property is not found in previous methods such as AccIHT[19].
>
> > I think X^TX takes O(n^2m) computational cost instead of O(mn)?
>
> A3. In the part of $X^T(X\theta^{t}-y)$ in Equation (2), we first compute $h=X\theta^{t}-y$ at $\mathcal{O}(mn)$ time, then compute $X^T h$ at $\mathcal{O}(mn)$ time. Consequently, Equation (2) requires $\mathcal{O}(mn)$ time. We would like to incorporate this explanation into our paper.
>
> > Did you mean to use f(\theta) instead of 1/2||y-X\theta||_2^2 in (1)?
>
> A4. Yes, for the sake of simplicity, we use the expression as stated in line 58.
>
> > I'd suggest the author add one or two sentences in the final Section to discuss the limitations of this work.
>
> We thank the reviewer for this comment. The limitation of our method is that the speedup factor decreases for large $k$ as described in line 270-272. However, as demonstrated in Theorem 1, the worst time complexity of our method is the same as that of the original IHT. We will incorporate this explanation into the paper.
>
> > To weaknesses:
>
> > It is unclear to me why the pruning strategy is defined the way it is.
>
> Kindly refer to our response in A2, which details the pruning strategy of our method and explains why this pruning strategy is safe. The point is that our method prunes unnecessary parameters before computing gradients by using upper bounds, which have small time complexity.
>
> > Consider adding more literature review: it is not clear how big of a gap exists in the literature that this work is trying to bridge. Section 4 Related Work feels out of place. Could consider moving this to the beginning of the paper.
>
> We would like to move Section 4 to the beginning of the paper. As explained in lines 245-246, while previous acceleration methods for IHT have reduced the number of iterations, there has not yet been a method to accelerate IHT by reducing the computation cost per iteration. This paper aims to fill this gap based on the pruning strategy.
>
> > A general suggestion: consider adding more explanation on the rationale behind each technical definition/ result and why it is defined/ stated in that particular way. For instance, it may be helpful to mention that Lemma 1 is derived from the triangle inequality + Holders inequality.
>
> We plan to explain the sketch of the proof and the reason for defining it before and after theorems and definitions, respectively.
>
> > I think some definitions are stated in the form of lemmas which they should not be, e.g. Lemmas 3 and 8. In general, I think some lemmas are unnecessary or can be combined.
>
> We will revise the manuscript to write Lemmas 3 and 8 as normal paragraphs or definitions.
>
> > Presentation is a bit long-winded at places, e.g. third paragraph
>
> We intend to shorten redundant expressions, such as those found in the third paragraph.
>
> > To minor comments:
>
> We plan to revise the paper by incorporating these helpful suggestions, including the notations.

---

> > ### Comment · Reviewer_abvp · 2024-08-08
> >
> > I thank the authors for their detailed response, in particular, for articulating the reason for choosing "pruning" over "thresholding", for explaining the gap in the literature that this work fills, and for agreeing to improve the structure and presentation of the paper. I decided to increase my rating to a weak accept.

---

> > > ### Author Response · Authors · 2024-08-09
> > >
> > > We would like to express our sincere gratitude for the response. We are currently revising the paper based on the feedback, and we believe that these changes have significantly improved its quality. If the reviewer has any further questions or suggestions, please do not hesitate to reach out to us.
> > >
> > > (Furthermore, the term "weak accept" in the reply is generally associated with a rating of 6 in this conference. We would greatly appreciate it if you could kindly verify whether the rating is accurate.)

---

> > > > ### Comment · Reviewer_abvp · 2024-08-09
> > > >
> > > > Thank you for pointing that out. I meant to say borderline accept, i.e. I've increased my rating by 2 points: one point to account for my own misunderstanding, and another to reflect the changes proposed by the authors. Overall I wanted to change my rating to indicate that this paper is worthy of publication.

---

> > > > > ### Author Response · Authors · 2024-08-09
> > > > >
> > > > > We thank the reviewer for verifying the rating. We greatly appreciate the positive feedback and will continue to work on improving the paper.

---

### Official Review · Reviewer_Lxri · 2024-07-11

**Soundness:** 2
**Presentation:** 2
**Contribution:** 2
**Rating:** 6
**Confidence:** 3

**Summary:**

The paper proposes to pruning the computation of marginal gradients in the IHT algorithm to accelerate the updating steps. For that, the upper bound $\overline{z}_j^{t}$ of the component $z_j^t$ in the gradient step is proposed in Definition 1 and  unnecessary elements in $\mathbf{z}^t$ that must be thresholded to zero can be identified.

**Strengths:**

- The proposed fast IHT is save in the sense that it can achieve the same output as original IHT.
- Pruning the unnecessary marginal gradient computation significantly saves computation costs when the sparsity $k$ is small. The idea is simple yet effective.
- Good empirical performance.

**Weaknesses:**

- The proposed upper/lower bounds seem to be very restricted to the structure of the sparse linear regression task. And the proposed method does not work when general convex loss functions are considered.
- For the sparse linear regression task, there are already efficient algorithms. For example, the cordinate descent algorithm uses in `glmnet`, which only needs to compute one dimension of marginal gradient in per iteration. The comparation beyond the IHT-type algorithm is ignored in the paper.

**Questions:**

There are some minor questions.
1. Line 42-45 is duplicated with the abstract.
2. Definition 2 is less informative. It generally says nothing about the construction of the candidate set $\mathbb{D}^t$.
3. The performance seems highly sensitive to the selection of $t^\ast$.
4. The proposed method seems only suitable when the learning rate $\eta$ is small and the parameters update gradually. When a larger learning rate is used or the momentum is introduced as in the related work (Section 4), the pruning may no longer take effect.

**Limitations:**

The limitation seems not addressed in the paper. My comments about it are already in the above sections of Weaknesses and Questions.

---

> ### Author Rebuttal · Authors · 2024-08-06
>
> We appreciate the reviewer’s constructive comments on our paper. To begin, we address the questions raised by the reviewer below.
>
> > 1. Line 42-45 is duplicated with the abstract.
>
> We thank the reviewer for this comment. We will revise those sentences accordingly.
>
> > 2. Definition 2 is less informative. It generally says nothing about the construction of the candidate set $\mathbb{D}^{t}$.
>
> We are grateful to the reviewer for this comment. We will revise the manuscript to write it as a paragraph rather than a definition.
>
> > 3. The performance seems highly sensitive to the selection of $t^*$.
>
> We greatly appreciate the reviewer for highlighting this critical point. The problem of sensitivity of $t^*$ is solved by the automatic determination of $t^*$ as described in lines 194-215. Specifically, the proposed method automatically adjusts $t^*$ during optimization to enhance performance, thereby eliminating the need to select $t^*$. According to Lemma 11, the error bounds of the upper and lower bounds may increase depending on the value of $t^*$, potentially reducing the pruning rate described in Definition 4 and consequently affecting performance. However, our method monitors the pruning rate and automatically adjusts $t^*$ to prevent a decrease in the pruning rate. Specifically, if the pruning rate decreases during optimization, the interval until the next $t^*$ is reduced following Equation (8). From Lemma 11, we know that the error bound of the upper and lower bounds tends to decrease when the interval until the next $t^*$ is small. Thus, this automatic adjustment of $t^*$ helps maintain the pruning rate.
>
> > 4. The proposed method seems only suitable when the learning rate is small and the parameters update gradually. When a larger learning rate is used or the momentum is introduced as in the related work (Section 4), the pruning may no longer take effect.
>
> Although the pruning rate might decrease due to the introduction of a large learning rate or momentum, the proposed method can adjust the pruning rate by automatically determining $t^*$ during optimization, as mentioned in our previous response.
>
> To demonstrate this, we conducted an additional experiment with a learning rate on the gisette dataset. The results are shown in PDF file in the global response. In the experiment, we increased the learning rate to 10 times that used in the experiment of Figure 1 (a) in our paper. The results exhibit a similar trend to Figure 1(a). Our method could accelerate IHT even with a large learning rate. We would like to highlight that all the objective values of AccIHT, which utilizes the momentum, diverged with the setting of $k=1280$, and we could not evaluate the processing times. This suggests that the method with momentum tends to fail when a large learning rate is used in our setting. In contrast, the methods without momentum, including our proposed method, converged in all settings. We would like to incorporate this finding into our paper.
>
> Finally, we would like to mention that even if the pruning rate of the proposed method is low, its worst time complexity remains the same as the original IHT (lines 218-224).
>
> > To Weaknesses:
>
> > The proposed upper/lower bounds seem to be very restricted to the structure of the sparse linear regression task. And the proposed method does not work when general convex loss functions are considered.
>
> As the reviewer pointed out, the proposed method is specialized for sparse linear regression with $\ell_0$ cardinality constraints. We agree that extending our method to general convex loss functions is important, and we plan to address this in future work. We would like to mention that the sparse linear regression with $\ell_0$ cardinality constraints is widely used in various fields, such as feature selection, sparse coding, dictionary learning, and compressed sensing, as described in lines 13--15. Although we performed a feature selection task in our experiment, we conducted an additional experiment on an image compressive sensing recovery task in response to Reviewer Hc1f. Our method completed the experiment in approximately 10 hours, while all other methods could not complete it within the author response period (4 days). We think that our method can contribute to reduce the times required for many tasks using IHT.
>
> > For the sparse linear regression task, there are already efficient algorithms. For example, the cordinate descent algorithm uses in glmnet, which only needs to compute one dimension of marginal gradient in per iteration. The comparation beyond the IHT-type algorithm is ignored in the paper.
>
> While it is true that there are many algorithms for accelerating sparse linear regression, the non-convexity due to $\ell_0$ cardinality constraints in Problem (1) of our paper limits the number of available acceleration methods, which is a crucial motivation for proposing our method. For instance, the coordinate descent method of glmnet, mentioned by the reviewer, is specialized for convex optimization with $\ell_1$ regularization, such as lasso and elastic net, and therefore cannot be used for the non-convex optimization with $\ell_0$ cardinality constraints in Problem (1). Similarly, screening [a] and working set algorithm [b] also accelerate sparse linear regression tasks but rely on the computation of a duality gap based on convex optimization and are not applicable to Problem (1). Although lines 245-252 of our paper contain a similar explanation, we believe that incorporating the above discussion will enhance the paper's readability, and we would like to reflect this in the revised manuscript.
>
> [a] O. Fercoq, A. Gramfort, J. Salmon, "Mind the duality gap: safer rules for the Lasso", ICML, 2015.
>
> [b] M. Massias, J. Salmon, and A. Gramfort, "Celer: a Fast Solver for the Lasso with Dual Extrapolation", ICML, 2018.
>
> > To Limitations:
>
> We intend to incorporate the above discussion into our paper. We sincerely appreciate the reviewer’s helpful comments.

---

> ### Comment · Reviewer_Lxri · 2024-08-10
>
> The authors' responses sufficiently address my questions. In response, I would like to raise my score. Overall, it is an interesting work from the computation aspect of the sparse regression problem.

---

> > ### Author Response · Authors · 2024-08-10
> >
> > We sincerely appreciate the reviewer's interest in our method. We are also deeply grateful for your assistance in improving the score. We will incorporate the valuable discussion above into our paper.

---

### Official Review · Reviewer_Hc1f · 2024-07-14

**Soundness:** 3
**Presentation:** 3
**Contribution:** 3
**Rating:** 5
**Confidence:** 2

**Summary:**

The authors accelerate the iterative hard thresholding (IHT) method, whose purpose is to find the k most important elements from a linear regression model. Specifically, they safely prune unnecessary elements with upper bounds on the element values. The experiment shows significant speedup for the proposed method.

**Strengths:**

The proposed method comes with theoretical guarantees and exhibit significant empirical speedup.

**Weaknesses:**

The importance of the work seems to be not clearly conveyed. The evaluation is also limited. While providing significant speedup, for the scale of the problem considered, the wall time of the baselines still seems to be within a tolerable range.

**Questions:**

Is it possible to include more experiments from the other use cases mentioned in the paper such as sparse coding, dictionary learning, or compressed sensing, preferably of a larger scale?

**Limitations:**

Yes, the authors discuss the limitations.

---

> ### Author Rebuttal · Authors · 2024-08-06
>
> We extend our sincere gratitude to the reviewer for working on our paper. We address the weaknesses and questions raised by the reviewer below.
>
> > The importance of the work seems to be not clearly conveyed.
>
> A1. As mentioned in lines 13—15, IHT is widely used in various fields, such as feature selection, sparse coding, dictionary learning, and compressed sensing. The original paper of IHT [6] has thereby been cited more than 2800 times. However, if design matrix $X$ is large, IHT suffers from high computation costs of gradient computations as described in 25—32. For instance, in our experiment of feature selection task, IHT requires over 24 hours for the total processing time, including hyperparameter search on the ledger and epsilon datasets, as shown in Figure 1 (c) and (e). Furthermore, in our answer of A3 below, we conducted an additional experiment on the image compressive sensing recovery task, and the baselines could not complete the task within 4 days. Our method completes all experiments, including additional experiments, within 24 hours, and we believe it can contribute to reduce the processing times required for many tasks using IHT.
>
> From a technical point of view, although accelerating IHT is the critical issue as described above, the number of approaches to accelerate IHT has been limited due to the non-convexity caused by $\ell_0$ cardinality constraints in Problem (1). As mentioned in lines 245-246, the previous approaches have reduced the number of iterations to accelerate IHT. In contrast, our approach reduces the computation cost per iteration by pruning unnecessary computations. Therefore, our method will provide a new direction for the acceleration of IHT. As illustrated in Figure 1, our approach significantly enhances the speedup factor compared to baselines using the previous approaches. We would like to incorporate this explanation into the beginning of our paper.
>
> > The evaluation is also limited. While providing significant speedup, for the scale of the problem considered, the wall time of the baselines still seems to be within a tolerable range.
>
> A2. We thank the reviewer for highlighting that issue. Regarding the comparison of processing times in Figure 1, the processing time for each $k$ and hyperparameter falls within a tolerable range. However, in our experiments, the total processing time for each baseline on the ledger and epsilon datasets exceeded 24 hours. While the acceptable range of processing time is subjective, many practitioners would likely find more than a day impractical. In contrast, our method took less than 24 hours, even for the ledger and epsilon datasets. We would like highlight that we conducted an additional experiment on the image compressive sensing recovery task, and the baselines could not complete the task within 4 days as we will explain in our answer of A3 below.
>
> > Is it possible to include more experiments from the other use cases mentioned in the paper such as sparse coding, dictionary learning, or compressed sensing, preferably of a larger scale?
>
> A3. We are grateful for the constructive feedback. We conducted an additional experiment on the image compressive sensing recovery task by following [a]. Although our experimental setting was almost the same as in Section 5 of [a], we used a large image with 2728$\times$2728 resolution. Consequently, the number of image blocks was 348,843. We investigated $k = [1, 8, 16, 24, 32]$ and evaluated the total processing times, including hyperparameter search. Our method completed the experiment in approximately 10 hours. Unfortunately, all other methods could not complete the experiment within the author's response period (4 days). The strong baselines of AccIHT and RegIHT also could not finish the experiment because they require additional hyperparameters for the weight step size and the momentum step size, respectively. We would like to highlight that our method does not require any additional hyperparameters, in contrast to AccIHT and RegIHT.
>
> [a] J. Zhang, C. Zhao, D. Zhao, and W. Gao, “Image compressive sensing recovery using adaptively learned sparsifying basis via L0 minimization”. Signal Process. 103: 114-126, 2014.

---

> > ### Comment · Reviewer_Hc1f · 2024-08-11
> >
> > Thank you for the responses.

---

> > > ### Author Response · Authors · 2024-08-12
> > >
> > > We sincerely appreciate the reviewer's response. If the reviewer has any further questions or suggestions, please feel free to reach out to us.

---

### Author Rebuttal · Authors · 2024-08-06

We would like to thank the reviewers for working on our paper. We have received many constructive comments, which we intend to incorporate to improve the quality of our work. We believe we have addressed most of the comments and would be grateful if the reviewers could respond to our response.

In this global response, we have attached a PDF file containing additional experimental results. Some reviewers suggested that our method might be slow with a large learning rate due to a potential decrease in the pruning rate. To address this concern, we conducted an additional experiment to evaluate the processing times with an increased learning rate.

In the experiment, we increased the learning rate to 10 times that used in Figure 1(a) of our paper. The results, detailed in the attached PDF, exhibit a similar trend to Figure 1(a). Our method was able to speed up IHT even with the larger learning rate. This success is due to our automatic determination of $t^{*}$, which adjusts the pruning rate during optimization, as described in lines 194—215. In contrast, AccIHT failed to converge in some cases, preventing us from evaluating the processing times for those instances. We would like to incorporate these findings into our paper.

---

### Decision · Program_Chairs · 2024-09-25

**Decision:**

Accept (poster)

**Comment:**

The paper proposes to pruning the computation of marginal gradients in the IHT algorithm to accelerate the updating steps. Experiemental results showed great speedup. Some reviewers raised scores after authors addressed review comments.